# Disparities in Cancer Screening Among the Foreign-Born Population in the United States: A Narrative Review

**DOI:** 10.3390/cancers17040576

**Published:** 2025-02-08

**Authors:** Andrew Rosowicz, Daniel Brock Hewitt

**Affiliations:** Department of Surgery, NYU Grossman School of Medicine, New York, NY 10016, USA; andrew.rosowicz@nyulangone.org

**Keywords:** foreign-born population, cancer screening disparities, colorectal cancer, cervical cancer, breast cancer, barriers to screening

## Abstract

Cancer screening is essential for early detection and improving survival rates; however, foreign-born individuals are significantly less likely to receive recommended screenings for colorectal, cervical, and breast cancer compared to U.S.-born individuals. This disparity is particularly concerning given the growing immigrant population, which now accounts for over 15% of the U.S. population. Many factors influence screening rates, such as length of residence, race and ethnicity, income, education, citizenship, insurance, usual source of care, language, medical literacy, and cultural barriers. Foreign-born immigrants face unique challenges, including limited access to public health insurance and fear of interacting with the healthcare system. This narrative review identifies the sociodemographic factors and barriers contributing to these disparities, and discusses interventions like education, patient navigation, and at-home testing to improve screening rates. Addressing these disparities is vital to ensure equitable healthcare access and improve cancer outcomes for this vulnerable and rapidly growing population.

## 1. Introduction

Between January 2021 and February 2024, the foreign-born population in the United States grew by 6.4 million, bringing the total number of foreign-born individuals to a record high of 51.4 million (Figure 1) [1]. This growth raised the foreign-born share of the U.S. population to 15.5%, the highest percentage recorded in U.S. history. According to the Census Bureau’s 2023 Current Population Survey, Latin America currently accounts for 54.0% of the foreign-born population, with Mexico alone representing 24.0%. Immigration from Latin America accounted for 4.1 million of the 6.4 million new immigrants from January 2021 to February 2024. During this period, South America and Central America experienced the largest growth in U.S. immigration, with increases of 49.4% and 36.8%, respectively. In this review, the term “immigrant” refers to any foreign-born individual residing in the United States, regardless of citizenship or insurance status. Foreign birth remains a significant barrier to cancer screening in the U.S. [2,3,4]. Policymakers must address this critical issue, given the growing number of immigrants and the increasing burden on the U.S. healthcare system.

The American Cancer Society (ACS) and the United States Preventive Services Task Force (USPSTF) provide clear screening guidelines for asymptomatic individuals at average risk of colorectal, cervical, and breast cancers [5,6,7]. The Centers for Disease Control and Prevention (CDC) reports a colorectal cancer screening rate of 72.9% among U.S. adults aged 50 to 75 [8]. For immigrants, the screening rate is significantly lower, with studies reporting a 49.6–52.8% completion rate [9,10,11,12]. For cervical cancer, 13.4–18.6% of foreign-born women report never having had a Pap test, compared to 5.2–6.8% of the U.S.-born population [13,14,15]. The CDC reports a rate of 76.7% for mammography completion among U.S.-born women, while the screening rate for immigrants is 65.5–68.5% [8,12,16]. Foreign-born individuals have lower screening rates for all three types of cancer, although there are nationwide screening recommendations for asymptomatic individuals at average risk.

One significant consequence of these screening disparities is later-stage diagnoses among immigrants. This trend has been consistently shown for colorectal, cervical, and breast cancer, and it remains true across different racial and ethnic groups [17,18,19,20,21,22,23,24]. Despite this finding, data on cancer survival and mortality are mixed. Some studies demonstrate worse outcomes among immigrants [24,25,26,27], while others paradoxically show improved survival, despite later-stage diagnoses [17,18,28,29]. One hypothesis that may explain this finding is the healthy migrant effect, which states that immigrants arrive at their destination country in a better state of health compared to domestic-born people [30]. Regardless, it is well established that later-stage diagnoses lead to worse outcomes, a trend consistently seen across multiple types of cancer [31,32,33].

This manuscript aims to describe the disparities in cancer screening for colorectal, cervical, and breast cancer among the U.S. foreign-born population. Specifically, this review will (1) characterize the sociodemographic factors associated with cancer screening, (2) examine barriers to cancer screening, and (3) evaluate interventions to improve cancer screening rates within this at-risk population.

## 2. Methods

This narrative review was conducted to examine disparities in cancer screening among the foreign-born population in the United States. Articles were identified through a systematic search of PubMed, using the keywords “cancer screening”, “immigrants”, “foreign-born”, and “United States”. The search was limited to studies published after 1 January 2000. Only studies related to colorectal, breast, and cervical cancer were included, as these are the only cancers with established screening recommendations for asymptomatic individuals at average risk. Limited data were available on other types of cancer screening among immigrants, reinforcing the decision to restrict the review to these three cancer types. Studies were also required to include quantitative data. After applying these criteria, a total of 42 articles were included in the final review (Figure 2). These studies provide a comprehensive overview of the disparities, barriers, and interventions related to cancer screening in immigrant populations.

The median publication year of the included studies was 2014, with five studies published in 2020 or later. Twenty-three studies were based on large nationwide or statewide surveys, with the National Health Interview Survey (NHIS) being the most commonly used. These studies had an average sample size of 4652 participants. The remaining 19 studies were smaller and more community-based, with an average sample size of 371 participants. Nearly half of the studies focused exclusively on women, and a similar proportion focused on a single racial or ethnic group. In terms of outcomes, most studies used USPSTF recommendations to determine whether immigrants had completed screening. Studies on interventions to increase screening frequently reported post-intervention knowledge scores, or whether screening was completed within a certain time frame after the intervention.

## 3. Factors Impacting Screening

### 3.1. Length of Residence in the United States

Length of residence in the U.S. significantly impacts cancer screening rates among immigrants. More recent immigrants (i.e., those who have spent fewer years residing in the U.S.) have lower screening rates for colorectal [4,9,11,12,34], cervical [4,12,13,14,15], and breast cancer [4,12,35,36]. These differences remain significant after adjusting for sociodemographic characteristics and access to care. A 2017 study including nearly 15,000 foreign-born individuals found that immigrants living in the U.S. for fewer than 5 years were nearly 70% less likely to be screened for colorectal cancer, and over 30% less likely to be screened for cervical and breast cancer [4]. More recently, a 2024 study similarly showed that immigrants living in the U.S. for over 15 years were 63% more likely to be screened for colorectal cancer [37]. Several studies focused on specific racial and ethnic groups demonstrate an association between duration of residence in the U.S. and cancer screening rates, specifically for African [38], Hispanic [39,40], and Asian immigrants [41,42,43].

Several explanations have been proposed for the association between length of residence and increased cancer screening among immigrants. Under the Patient Protection and Affordable Care Act passed in 2010, immigrants become eligible for public health insurance after five years of legal residence in the U.S.; however, this is unlikely to completely account for this association, given that studies control for insurance status and other barriers to screening, such as citizenship and having a usual source of care. Another explanation is that over time, immigrants become acculturated, gaining a better understanding of the importance of cancer screening and overcoming cultural barriers such as fear of testing and deterministic or fatalistic beliefs regarding cancer outcomes. Overall, duration of residence in the U.S. may serve as a proxy for acculturation, with longer residency associated with higher cancer screening rates.

### 3.2. Race, Ethnicity, and Country of Origin

Cancer screening rates differ among racial and ethnic immigrant populations. Similarly to the U.S.-born population, Hispanic and Asian immigrants have lower colorectal cancer screening rates compared to the overall immigrant population, after adjusting for sociodemographic characteristics and access to care (Table 1) [4,8,9,10,11,44]. No significant differences are observed for other racial and ethnic groups. Cervical cancer screening rates among immigrants also follow trends observed in the U.S.-born population, with African immigrants having the highest rates, and Asian immigrants having the lowest [4,8,15]. Among the U.S.-born population, breast cancer screening rates are highest among African Americans and lowest among Hispanics [8]. African immigrants have the highest screening rates among the foreign-born population, while Asian immigrants have the lowest [4,16,35]. Overall, Asian immigrants exhibit the lowest screening rates across all cancer types, while African immigrants have the highest. Given the large number of Asian immigrants in the U.S. and their diverse countries of origin, targeted efforts are needed to address screening disparities in this group.

Several studies have also investigated differences in screening based on immigrant country of origin. Among Asian immigrants, Filipinas exhibit higher screening rates compared to other Asian subgroups [41,45]. Dominicans have the highest screening rates among Hispanic immigrants, while immigrants from Mexico, which represents the largest single country of origin among U.S. immigrants, are the least adherent [36,39,46,47,48]. Interestingly, screening differences are inconsistent across all cancer types. For example, a 2014 study focusing on subgroups of African immigrants found that Somali immigrants were less likely to have ever had a Pap test compared to other African immigrants, but were more likely to have undergone a mammogram [38]. These variations within racial and ethnic groups and across cancer types underscore the complexity of sociocultural factors influencing screening behaviors.

### 3.3. Specific Cancer Screening

Screening rates differ between types of cancer for both the U.S.-born and immigrant populations. In the U.S., screening rates for colorectal, cervical, and breast cancer in 2021 were 72.2%, 75.2%, and 75.7%, respectively [49]. Studies on immigrants consistently show that cervical cancer screening rates are the highest, followed by breast and then colorectal cancer screening [3,12,34]. Differences in screening rates are much greater for immigrants compared to U.S.-born individuals. A 2017 study found that the colorectal cancer screening rate for immigrants who had lived in the U.S. for over 10 years was 52.3%, compared to rates of 79.3% and 70.0% for cervical and breast cancer [12].

Cancer screening rates for immigrants are associated with the level of coordination required to complete each test. Immigrants are most adherent to cervical cancer screening, likely because Pap tests can be performed as in-office procedures by multiple different care providers. Despite being less invasive, mammograms typically require a separate visit to an imaging center after referral. Finally, colorectal cancer screening, which is most often completed by colonoscopy, is the most invasive, and requires the most coordination and preparation beforehand. U.S.-born adults also have a lower rate of colorectal cancer screening compared to cervical and breast cancer screening; however, this difference is much greater for immigrants, who face additional barriers to cancer screening. Differences in screening are likely unrelated to the curability of each cancer type, as colorectal, cervical, and breast cancer all exhibit similarly high survival rates when diagnosed at an early stage.

### 3.4. Income and Education Level

Cancer screening among immigrants differs significantly by level of income. In 2021, the poverty rate for foreign-born households was 14.8%, compared to 11.6% for U.S.-born households; furthermore, this figure reached 20% for undocumented immigrants [50]. A 2022 study found that immigrants with an income level at least double the poverty threshold were nearly 50% more likely to be screened for colorectal cancer compared to immigrants living below the poverty threshold [10]. Several studies have shown stepwise increases in rates of screening with increasing levels of income [4,12,37,44,51]. Immigrants with an income of $75,000 or greater were found to be nearly 80% more likely to be screened compared to those with an income below $20,000 [44,51]. These associations remained significant after adjusting for sociodemographic characteristics and access to care. The trend was also found to be consistent across different racial and ethnic groups [38,52]. Overall, a stepwise relationship has been observed between cancer screening rates and income level, with lower-income immigrants having lower screening rates. In addition to having limited access to screening resources, immigrants with lower income may also perceive that they would lack the financial means to afford treatment if they were to receive a positive diagnosis.

Education has been similarly associated with cancer screening among immigrants, with several studies showing a stepwise relationship between education level and cancer screening rates [4,10,12,15,37,44]. Compared to patients with a college degree or higher, patients with a high school degree were found to be 10–34% less likely to be screened, and patients with less than a high school degree were found to be 26–43% less likely to be screened [4,44]. Consistently across different types of cancer screening, immigrants with lower levels of education were found to have significantly lower screening rates.

### 3.5. The COVID-19 Pandemic

The COVID-19 pandemic significantly disrupted cancer screening rates among both U.S.-born and foreign-born individuals, driven by the reallocation of healthcare resources away from preventive services, patients’ reluctance to visit medical facilities due to fear of infection, and widespread staffing shortages. Among U.S.-born populations, screening rates for colorectal, cervical, and breast cancer declined by 35–45% in the first half of 2020, and these rates have yet to fully recover [53]. While data on the foreign-born population are limited, a U.S.-based 2023 study reported a significant decline in cervical cancer screening among immigrants between 2019 and 2021 [54]. A 2023 study in Denmark similarly showed that immigrants experienced disproportionate reductions in cervical cancer screenings during the pandemic [55]. In contrast to cervical cancer, the U.S. study revealed an increase in colorectal cancer screening among immigrants, driven largely by a surge in the use of stool-based testing [54]. This increase in stool-based testing was similarly observed for U.S.-born individuals [56]. In addition, a separate study in Denmark found that immigrants were among the groups with the highest uptake of stool testing during the pandemic [57]. Separately, a 2023 study in Ontario, Canada found that the COVID-19 pandemic exacerbated disparities in colorectal and breast cancer screening among immigrants, whereas cervical cancer screening rates remained unaffected [58]. The pandemic’s impact on cancer screening among immigrants varied across countries and type of cancer screening. The increased utilization of stool testing during the pandemic, however, highlights a potential opportunity to address screening disparities among the foreign-born population.

## 4. Barriers to Screening

### 4.1. Citizenship

Lack of U.S. citizenship or permanent residency is a significant barrier to care that many immigrants face after arriving in the U.S., and it is the only barrier that exclusively applies to foreign-born individuals. Among the U.S. foreign-born population, less than half are naturalized citizens, and nearly one-quarter are undocumented migrants [59]. A 2015 study examining all three types of cancer screening found that the disparity between citizens versus non-citizens was most pronounced for colorectal cancer, while differences in cervical and breast cancer screening were minimal [3].

Several studies have identified citizenship status as an independent predictor of colorectal cancer screening among immigrants (Table 2) [9,10,11,37]. Citizenship influences screening by enabling access to public health insurance; however, these studies show that immigrant non-citizens still have lower screening rates even after adjusting for insurance status [10,11]. Citizenship, therefore, likely impacts colorectal cancer screening through additional pathways other than providing access to health insurance. For example, citizenship may reduce fear of discrimination or deportation, leading to greater engagement with the healthcare system and uptake of preventative services [9].

In contrast to colorectal cancer, multiple studies on breast cancer screening report no differences based on citizenship status [16,45,60]. A 2011 study found lower breast cancer screening rates among non-citizen immigrants after adjusting for sociodemographic factors; however, this disparity disappeared after accounting for insurance status and access to a usual source of care [60]. Data on cervical cancer screening are limited; however, available evidence and overall trends for different types of cancer screening suggest that there is no significant difference between immigrant citizens and non-citizens [3]. Given that colorectal cancer screening is the most invasive type of screening and requires the most coordination, factors such as lack of knowledge, inability to coordinate appointments, cultural beliefs, or fear of screening procedures likely play a larger role in screening behaviors.

### 4.2. Health Insurance Status

Insurance status is strongly associated with screening rates, and is one of the most significant barriers to cancer screening for immigrants. The uninsured rate for the U.S.-born population is 8%, compared to 15% for immigrants [62]. Within the immigrant population, 50% of undocumented immigrants do not have health insurance. Immigrant non-citizens become eligible for public health insurance after five years of legal residence under the Affordable Care Act (ACA). Since nearly 30% of immigrants fall into this category of legal non-citizens, the ACA was a significant step toward increasing health insurance access within the foreign-born population [59]. Colorectal, cervical, and breast cancer all have significant differences in screening rates based on insurance status (Table 2) [3,4,10,11,12,16,37,42].

Studies examining all three types of cancer screening generally find the largest difference in screening rates for colorectal cancer, followed closely by breast cancer [3,4,12,42]. Multiple studies on colorectal and breast cancer show that insured immigrants are at least twice as likely to complete screening as those without insurance [4,10,11,16,42]. These models are adjusted for sociodemographic characteristics and other common barriers to screening, such as citizenship and usual source of care. A single study comparing cervical cancer screening between insured and uninsured immigrants found significantly higher rates of screening for the insured group, based on univariate analysis [15]. A 2017 study also compared private versus public health insurance, and found that patients with public insurance were 15% less likely to complete screening [4].

### 4.3. Usual Source of Care

Along with being uninsured, lacking a usual source of care is one of the most significant barriers to cancer screening for immigrants. A usual source of care refers to a regular physician or a community-based clinic where an individual routinely seeks medical services. Nearly 20% of immigrants do not have a usual place of care, compared to 12% of U.S.-born individuals [62,63]. This percentage rises to 38% among undocumented migrants and 42% among uninsured immigrants. Immigrants with no usual source of care have lower screening rates for colorectal, cervical, and breast cancer (Table 2) [4,11,12,15,16,37,42,61].

Studies examining all three types of cancer screening show mixed results regarding which type is most impacted. Similarly to health insurance, having a usual source of care is significantly associated with higher screening rates, after controlling for other factors and barriers to screening [4,11,16,42,61]. Studies examining both insurance status and usual source of care find similar degrees of influence. Immigrants with health insurance or a usual source of care are more than twice as likely to complete cancer screening [4,11,16,42]. Multivariable models suggest that having a usual source of care enables immigrants to complete screening, regardless of insurance status. Providers who regularly work with immigrants are likely equipped to help them to access alternative methods for obtaining screening without relying on insurance.

### 4.4. Language Barrier

Among the U.S. immigrant population, 47% of individuals report being able to speak English “very well ” [62]. Over 80% of immigrants speak a language other than English at home, and half speak Spanish. Many studies examine the ability to speak English as a barrier to care, and the majority find no difference in screening between English-speaking and non-English-speaking immigrants [10,42,45,60]. These studies all control for sociodemographic characteristics and access to care. While several groups do report lower screening rates for non-English-speaking immigrants [40,44,64], there are just as many that report higher screening rates for non-English speakers [46,52,61]. Additionally, studies reporting lower screening rates for non-English speakers are generally older, and were possibly conducted before over-the-phone and video interpreters were as readily available. Two studies took place in the context of a colonoscopy referral and patient navigator program, and interestingly, both report that Spanish-speaking immigrants were twice as likely to receive screening after these interventions [46,52]. Immigrants who are less acculturated to the U.S. may be more open to receiving screening through such programs. Overall, English-speaking ability is not a significant barrier to cancer screening. Recent studies report no difference in screening rates, likely due to co-linearity with other sociodemographic factors and the expansion of translator services.

### 4.5. Medical Literacy and Cultural Barriers

Because medical literacy and cultural factors are difficult to quantify, there are limited data on these barriers to screening. A 2011 study assessed medical literacy using a questionnaire on breast and cervical cancer, finding that Hispanic women with higher knowledge scores were more likely to be adherent to mammography and Pap testing [39]. Although quantitative data on medical literacy are scarce, many interventions to increase cancer screening among immigrants emphasize education.

Cultural barriers to cancer screening include fatalism, beliefs about illness, fear of screening procedures, and fear of a cancer diagnosis. Fatalism, the belief that events are inevitable and cannot be changed, is common among immigrants, and has been identified as a barrier to screening across multiple racial and ethnic groups [65,66]. The impact of cultural factors is most often captured by smaller qualitative studies on specific immigrant communities. Meta-analyses synthesizing these studies identify cultural factors as significant barriers to colorectal, cervical, and breast cancer screening [67,68,69]. Consequently, many interventions aimed at increasing screening among immigrants are culturally tailored to specific groups, presenting information and guidance in ways that are more accessible.

## 5. Interventions

### 5.1. Education

Education is a key strategy for improving cancer screening rates among both immigrants and the U.S.-born population. Educational seminars on cancer screening cover cancer incidence, risk factors, screening processes, benefits, and overcoming barriers. Existing studies on immigrants have small sample sizes and provide limited evidence on whether education directly increases screening rates. For instance, a 2015 study of Chinese immigrant women in Portland found that individual counseling sessions significantly improved breast cancer and mammography knowledge, with sustained effects over 12 months [70]. Despite this positive result, the study did not evaluate actual screening behaviors. Similarly, another 2015 study reported increased knowledge scores after immigrants participated in focus group sessions, but found no significant change in Pap test uptake [71]. A 2015 study on Iraqi women found that older women were less likely to show improved knowledge scores following an educational session, due to a resurgence of fear and anxiety [72]. The only study showing a direct effect on screening focused on Korean women in Alabama [73]. Participants who attended four weekly, hour-long education sessions showed an increased rate of Pap testing within three months of intervention. This study notably did not report any long-term outcomes. Although community-level educational interventions improve knowledge about cancer screening, evidence of their impact on actual screening rates among immigrants remains limited.

### 5.2. Patient Navigation

Immigrants often face challenges in navigating the healthcare system in a foreign country, compounded by significant barriers to care. Patient navigators support immigrants in accessing cancer screening by scheduling appointments and guiding them through the healthcare system and insurance processes. Similarly to educational interventions, data on the effectiveness of patient navigators for improving cancer screening among immigrants are limited in terms of the number and size of studies. Two studies conducted within a community health clinic affiliated with Massachusetts General Hospital demonstrated that patient navigators effectively increased breast cancer screening rates among foreign-born women. In one study, breast cancer screening rates for Bosnian-speaking women increased by 23% within one year of the intervention [74]. Another study reported a similar increase in screening among all refugee women over three years [75]. A 2014 study on colorectal cancer screening in Hispanic immigrants interestingly found no difference in screening rates between culturally targeted patient navigation and standard patient navigation [52]. These studies suggest that patient navigation is effective in helping immigrants to overcome barriers to screening, even without a culturally tailored approach.

Of note, two studies examined the combined effects of patient education and navigation on cancer screening among immigrants. In one study, Hispanic immigrants were provided with a colorectal cancer educational video, a brochure highlighting key information, and a form for their primary care physician to order screening. The intervention group demonstrated a significantly higher colorectal cancer screening completion rate compared to the control group (55% vs. 18%) [76]. In a separate study, Korean women who received multimedia messages through a mobile phone app, in addition to navigator services, had significantly higher knowledge scores and screening compared to women who received a printed brochure alone (75% vs. 30%) [77]. Although data remain limited, small community-based studies indicate that patient navigation is effective in increasing cancer screening rates among immigrants. The combined intervention studies showed a significant increase in screening, indicating a potential synergistic effect between education and patient navigation.

### 5.3. At-Home Testing

At-home testing is another strategy to improve screening rates among both immigrants and the U.S.-born population for cancers where non-invasive methods are effective. Among the U.S.-born population, at-home FIT tests and HPV tests have been shown to increase participation when mailed to patients overdue for screening [78,79]. A 2015 study offered at-home HPV testing to Somali women who had not undergone a Pap test in the previous three years, and found that completion rates for at-home testing were 14 times higher than for clinic-based Pap tests [80]. This is the only quantitative study that has examined at-home testing within the immigrant population, underscoring the need for more data on this intervention. While at-home tests improve access to cancer screening, there is a risk of incorrect sample collection, particularly among immigrants who may have limited English proficiency or low health literacy.

Education, patient navigators, and at-home testing have all shown the potential to increase cancer screening rates among immigrants. Although data specific to immigrants are limited, meta-analyses conducted on the U.S.-born population demonstrate the consistent effectiveness of these interventions [81,82,83,84]. Preliminary findings suggest that patient navigation and at-home testing may be particularly effective for immigrants, as these strategies directly address the significant barriers to care that this population often encounters.

## 6. Limitations

This review has several limitations. There has been a significant increase in immigration to the United States in recent years; however, only five studies published since 2020 were identified for this review. Additionally, there were few studies which examined the impact of the COVID-19 pandemic on cancer screening among immigrants. This review focused on colorectal, cervical, and breast cancer, as these diseases have more congruent screening guidelines for asymptomatic individuals at average risk. While this was a necessary limitation for the scope of this review, this choice excluded other types of cancers that may also exhibit screening disparities. A significant number of qualitative studies were also excluded. These were mostly small studies based on interviews with local immigrant communities. While these studies offer valuable insight into the cultural barriers to screening that immigrants face, there is significant heterogeneity, which limits the generalizability of their findings. Furthermore, there were no available data comparing screening rates for immigrants from different states or regions of the country. Because there have been no interventions on a national scale, data on interventions to increase cancer screening among immigrants are also limited to small, community-based studies. In this review, the average sample size for these studies was only 137 participants. Overall, these limitations highlight the need for more large-scale, national studies to better understand and address the barriers to cancer screening among immigrant populations.

## 7. Conclusions

A complex interplay of different factors and barriers to screening shapes cancer screening disparities among U.S. immigrants. Overall, screening rates for colorectal, cervical, and breast cancer are significantly lower among immigrants compared to the U.S.-born population. Within the immigrant population, rates are influenced by length of residence in the U.S., race and ethnicity, specific type of cancer screening, income and education level, citizenship, health insurance, and having a usual source of care. Analyzing these factors has allowed us to identify immigrants at the highest risk of being unscreened, and enables targeted intervention. Interventions such as education, patient navigators, and at-home testing have shown promise in increasing cancer screening rates among immigrants. With the significant growth in the immigrant population in recent years, addressing these disparities is essential for improving early cancer detection and reducing mortality among immigrants.

## 8. Future Directions

Given the recent influx of immigrants to the U.S., older studies do not adequately reflect the current immigrant population. It is essential to continue investigating the factors and barriers that influence screening, in order to better understand the unique challenges faced by this evolving demographic. In particular, data are needed on how the COVID-19 pandemic has specifically affected cancer screening among foreign-born individuals, given the known detriments that COVID-19 had on cancer screening among all populations. In addition, more research is required to explore the consequences of these screening disparities. To effectively advocate for interventions that promote increased screening among immigrants, stronger evidence is needed to demonstrate that decreased cancer screening in this group leads to worse health outcomes.

Interventions aimed at increasing cancer screening among immigrants should be implemented at the state and city levels, given that a relatively small number of cities and states account for the majority of the nation’s foreign-born population [85]. Based on the findings of this review, a lack of a usual source of care is the most significant barrier to screening for immigrants, even more so than a lack of health insurance. Establishing more clinics that serve immigrants regardless of citizenship or insurance status would substantially increase screening rates. Most clinics currently dedicated to providing care for immigrants are small, community-run clinics with limited resources. There is a pressing need for greater resource allocation and coordinated efforts on a broader scale. At-home testing is another promising strategy to increase screening while reducing the burden on both foreign-born individuals and the healthcare system. This form of screening gained popularity during the COVID-19 pandemic and led to increased colorectal cancer screening among the U.S. immigrant population [54]. Proven to be effective among vulnerable U.S.-born individuals, at-home testing should be expanded to the foreign-born population. Education and patient navigation are two interventions that have worked at the individual patient level. While education alone has been shown to have limited success in increasing screening, combining education with patient navigation has been shown to be highly effective [76,77]. Education is necessary to demonstrate the benefits of screening and overcome cultural barriers, while patient navigators help immigrants to navigate the complexities of the healthcare system and access screening services. To achieve more impactful results, future interventions should integrate these two strategies.

Emerging technologies, such as artificial intelligence (AI), offer new opportunities to overcome barriers to cancer screening for immigrants. AI has been applied to improve the accuracy of screening tests for colorectal [86], cervical [87], and breast cancer [88], which is particularly beneficial for immigrants who have limited access to screening services. One advantage of AI-assisted mammograms is the provision of immediate results, enabling immigrants to receive faster notification of a positive finding, and reducing the risk of being lost to follow-up [89]. Additionally, AI has been used to identify individuals at increased risk of developing cancer, a tool that could be extended to immigrant populations to target high-risk individuals effectively [90]. Generative AI also has the potential to create personalized educational materials to inform immigrants about the importance of cancer screening and the procedures involved. Furthermore, AI could facilitate individualized patient navigation by helping immigrants to locate resources, schedule appointments, and communicate results, thereby addressing key barriers to cancer screening access and adherence. More exploration is needed in this area to fully understand and harness AI’s potential to improve cancer screening for immigrant populations.

## Figures and Tables

**Figure 1 cancers-17-00576-f001:**
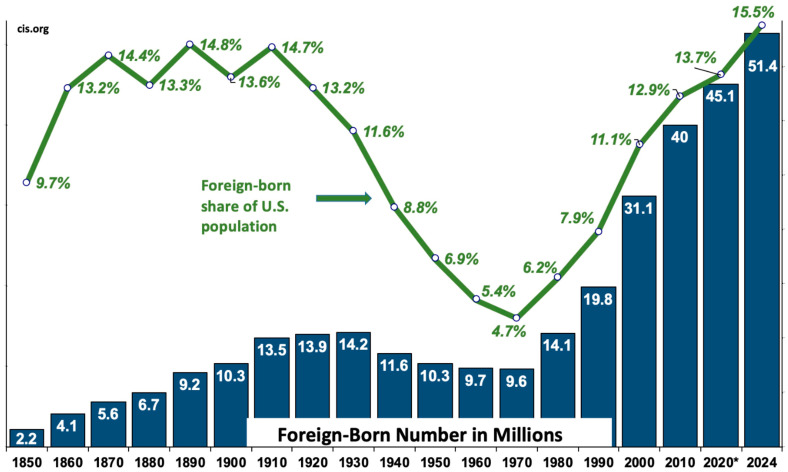
Growth of the United States’ foreign-born population over time. Sources: Decennial Census for 1850 to 2000, American Community Survey for 2010 and 2020, and February current population survey for 2024. * The 2020 ACS shows only 43.5 million foreign-born residents—13.2% of the population; however, the Census Bureau reports that it does not have confidence in the 2020 ACS due to pandemic related issues. Averaging the 2019 and 2021 ACS shows a foreign-born number that was 45.1 million and 13.7% of the population.

**Figure 2 cancers-17-00576-f002:**
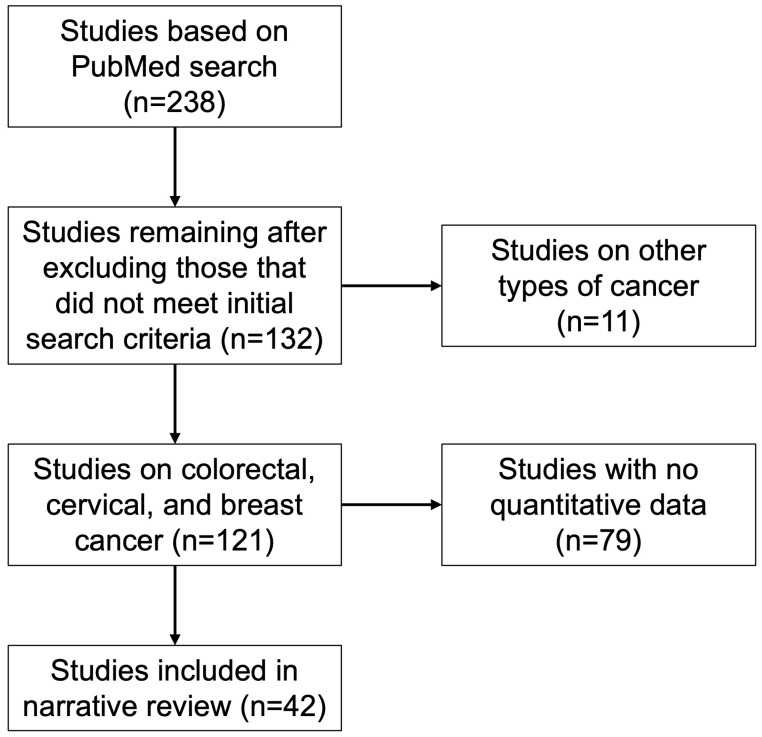
PubMed search and study selection criteria.

**Table 1 cancers-17-00576-t001:** Adjusted odds ratios for cancer screening among foreign-born individuals.

	White	Black	Hispanic	Asian
Colorectal Cancer				
Okitondo et al. (2024) [37]	Ref	1.01 (0.80–1.29)	0.66 (0.55–0.80) *	0.83 (0.68–1.00) *
Yao et al. (2022) [10]	Ref	0.93	0.91	0.63 *
Cofie et al. (2020) [11]	Ref	0.91 (0.71–1.17)	0.87 (0.72–1.06)	0.63 (0.52–0.76) *
Miranda et al. (2017) [4]	Ref	1.05 (0.98–1.14)	0.75 (0.66–0.85) *	0.61 (0.51–0.72) *
Liss et al. (2014) [44]	Ref	1.02 (1.00–1.04)	0.94 (0.91–0.98) *	0.76 (0.69–0.83) *
Cervical Cancer				
Miranda et al. (2017) [4]	Ref	1.85 (1.72–1.99) *	1.52 (1.35–1.71) *	0.55 (0.47–0.65) *
Breast Cancer				
Miranda et al. (2017) [4]	Ref	1.47 (1.37–1.59) *	0.85 (0.71–1.01)	0.65 (0.55–0.75) *
Yao et al. (2014) [16]	0.77 (0.47–1.28)	0.93 (0.41–2.11)	Ref	0.74 (0.45–1.22)

Odds ratios adjusted for sociodemographic characteristics (age, race/ethnicity, marital status, years in the U.S., income level, education level) and access to care (citizenship, health insurance, usual source of care, health status). * *p* < 0.05.

**Table 2 cancers-17-00576-t002:** Adjusted odds ratios for barriers to cancer screening among foreign-born individuals.

	Population	Type of Cancer	Odds Ratio
**Citizenship**			
Okitondo et al. (2024) [37]	All immigrants	Colorectal cancer	1.39 (1.17–1.65) *
Yao et al. (2022) [10]	All immigrants	Colorectal cancer	1.54 *
Cofie et al. (2020) [11]	All immigrants	Colorectal cancer	1.35 (1.14–1.60) *
Yao et al. (2014) [16]	All immigrants	Breast cancer	1.18 (0.75–1.87)
Ryu et al. (2013) [45]	Asian immigrants	Breast cancer	1.65 (0.69–3.93)
Billmeier et al. (2011) [60]	All immigrants	Breast cancer	1.33 (0.76–2.33)
**Health Insurance**			
Okitondo et al. (2024) [37]	All immigrants	Colorectal cancer	2.22 (1.73–2.86) *
Yao et al. (2022) [10]	All immigrants	Colorectal cancer	2.70 *
Cofie et al. (2020) [11]	All immigrants	Colorectal cancer	2.38 (1.90–2.99) *
Miranda et al. (2017) [4]	All immigrants	Colorectal cancer	2.33 (2.08–2.63) *
		Cervical cancer	2.33 (2.17–2.50) *
		Breast cancer	2.56 (2.38–2.78) *
Lee et al. (2014) [42]	Asian immigrants	Colorectal cancer	2.15 (1.30–3.56) *
		Cervical cancer	1.70 (1.02–1.85) *
		Breast cancer	2.27 (1.40–3.69) *
Yao et al. (2014) [16]	All immigrants	Breast cancer	2.56 (1.41–4.64) *
Ryu et al. (2013) [45]	Asian immigrants	Breast cancer	2.42 (0.88–6.64)
**Usual Source of Care**			
Okitondo et al. (2024) [37]	All immigrants	Colorectal cancer	2.91 (2.21–3.82) *
Cofie et al. (2020) [11]	All immigrants	Colorectal cancer	3.63 (2.85–4.62) *
Miranda et al. (2017) [4]	All immigrants	Colorectal cancer	2.86 (2.63–3.13) *
		Cervical cancer	2.17 (2.08–2.33) *
		Breast cancer	3.13 (2.86–3.33) *
Lee et al. (2014) [42]	Asian immigrants	Colorectal cancer	1.79 (1.11–2.87) *
		Cervical cancer	2.60 (1.60–4.22) *
		Breast cancer	1.82 (0.83–3.98)
Yao et al. (2014) [16]	All immigrants	Breast cancer	2.22 (1.10–4.46) *
Ellison et al. (2011) [61]	Hispanic immigrants	Colorectal cancer	3.56 (1.04–12.13) *

Odds ratios adjusted for sociodemographic characteristics (age, race/ethnicity, marital status, years in the U.S., income level, education level). Reference groups for each barrier to care were undocumented individuals/those with unauthorized status, uninsured individuals, and individuals lacking a usual source of care, respectively. * *p* < 0.05.

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
