# Peer review of "Disparities in Cancer Screening Among the Foreign-Born Population in the United States: A Narrative Review"

_cancers, 2025, doi:10.3390/cancers17040576_

Round 1

Reviewer 1 Report

Comments and Suggestions for Authors

Thank you for the opportunity to review manuscript ID: cancers-3432070. This narrative review aimed to describe the disparities in cancer screening for colorectal, cervical, and breast cancer among the U.S. foreign-born population, as well as to explore the sociodemographic factors, barriers, and interventions influencing cancer screening rates among foreign-born individuals in the U.S.

Comments:

The applied methodology is not described in this manuscript (narrative review). Correct this, citing appropriate references.

Add a new section (Methods) in which the following should be specified:

Describe the rigor/transparency of search methods in detail.

The authors should indicate which databases they searched and which search terms were included. This article does not provide the exact search strings used for the different databases nor the number of results for different searches. Including the number of search results would be particularly helpful in contextualizing the studies that were ultimately included in the review and increase the rigor and transparency of the article.

Specify selection criteria.

Specify the time frame covered by this narrative review. 

Pay special attention to the period of the COVID-19 pandemic and the effect on cancer screening.

How might the heterogeneity of the reviewed studies be discussed?

Before the Conclusions section, add a new paragraph in which it is mandatory to state the limitations of this manuscript, with a discussion of the possibilities for overcoming the shortcomings.

The Tables suggests the reviewed studies did not investigate other experiences (the practice of cancer screening during the COVID-19 pandemic) and findings related to these characteristics or experiences could also be taken up in the discussion, where they are currently not given space.

A significant number of cited references were published more than 10 years ago. Correct this, in such a way that the old references are replaced by the recent appropriate references wherever possible.  

Reviewer 2 Report

Comments and Suggestions for Authors

In this narrative review, " Disparities in Cancer Screening among the Foreign-Born Population in the United States: A Narrative Review," the authors discussed about the cancer care disparities between native American citizens and immigrants. Although the subject is well taken and needs to be discussed, it must be presented with more depth and understanding. These are some of the concerns raised :

1. The conclusion and discussion section are drafted quite poorly and don't justify the weight of the manuscript. The manuscript is more about raising concerns about the system without offering an actionable solution. Please re-write and elaborate the discussion section thoroughly to incorporate possible remedies.

2. There is no quantitative representation showing the effect of income disparities between immigrants and Americans and how this could change the healthcare landscape

3. The authors mentioned that "Immigrants have lower screening rates for colorectal, cervical, and breast cancer compared to U.S.-born individuals, with the largest disparities observed in colorectal cancer." No reason was given why there are cancer-specific disparities and how they may vary from state to state. Please also refer to whether this may be due to the availability of curative interventions. Or affordability of medicines and surgery? The availability of insurance alone cannot explain these cancer-specific disparities. Then why less in breast cancer?

4. The authors mention that ". Overall, English-speaking ability is not a significant barrier to cancer screening" Line 247. Logically, this seems to be a significant player due to the lack of proper communication between patients and healthcare providers. The patient navigation section seems to prove this same point. Justify

5. Section 4:3 . At-home testing is not a logical solution, as many cancer screenings need hospital visits like doing mammography or a colonoscopy. Why would that be significant?

The overall writeup is good and clearly presented, though readers expect to see more visual representations like graphs, bars, and pie charts. Authors should try some graphical representations.

Reviewer 3 Report

Comments and Suggestions for Authors

The authors aimed to explore the sociodemographic factors, barriers, and interventions influencing cancer screening rates among foreign-born individuals in the U.S. The topic is valuable due to the higher presence of discrepancy in access to screening programs within race/ethnicity minorities. However, several improvements should be addressed. First, Table 1 should be better discussed and the caption may be improved. Reporting OR above and below the 1 sounds misleading. Which race/ethnicity groups have the disadvantage in access to treatment? However, Table 1 should be replaced. Indeed, Table 1 should display ORs in predicting access to screening was tested. Second, why prostate cancer was not addressed? It is one of the major cancers that is suitable for screening (PMID 38051582). Table 2 should be also re-edited. What were the barriers that reduced the access to screening for these race/ethnicity minorities? Any data on survival outcomes (PMID 38509444)? 

Round 2

Reviewer 1 Report

Comments and Suggestions for Authors

Thank you for the opportunity to re-review manuscript ID: cancers-3432070.

The authors have correctly addressed all my comments (point by point) and made appropriate corrections in the revised version of this paper. I believe that this paper is now much clearer and more informative for everyone dealing with this topic. I thank the authors for their efforts in revising this paper. 

Reviewer 2 Report

Comments and Suggestions for Authors

The revised version of the manuscript is substantially improved and maybe accepted

Reviewer 3 Report

Comments and Suggestions for Authors

No further comments are needed